# Expression Profile and Biological Role of Immune Checkpoints in Disease Progression of HIV/SIV Infection

**DOI:** 10.3390/v14030581

**Published:** 2022-03-11

**Authors:** Yuting Sun, Jing Xue

**Affiliations:** 1Beijing Key Laboratory for Animal Models of Emerging and Re-Emerging Infectious Diseases, Institute of Laboratory Animal Science, Chinese Academy of Medical Sciences, Beijing 100021, China; sunyuting9077@163.com; 2NHC Key Laboratory of Human Disease Comparative Medicine, Comparative Medicine Center, Peking Union Medical College, Beijing 100021, China; 3Center for Acquired ImmunoDeficiency Syndrome Research, Chinese Academy of Medical Sciences and Peking Union Medical College, Beijing 100730, China

**Keywords:** human immunodeficiency virus (HIV), simian immunodeficiency virus (SIV), immune checkpoint, immune checkpoint blocker, HIV therapy

## Abstract

During HIV/SIV infection, the upregulation of immune checkpoint (IC) markers, programmed cell death protein-1 (PD-1), cytotoxic T-lymphocyte-associated antigen-4 (CTLA-4), T cell immunoglobulin and ITIM domain (TIGIT), lymphocyte-activation gene-3 (LAG-3), T cell immunoglobulin and mucin domain-3 (Tim-3), CD160, 2B4 (CD244), and V-domain Ig suppressor of T cell activation (VISTA), can lead to chronic T cell exhaustion. These ICs play predominant roles in regulating the progression of HIV/SIV infection by mediating T cell responses as well as enriching latent viral reservoirs. It has been demonstrated that enhanced expression of ICs on CD4^+^ and CD8^+^ T cells could inhibit cell proliferation and cytokine production. Overexpression of ICs on CD4^+^ T cells could also format and prolong HIV/SIV persistence. IC blockers have shown promising clinical results in HIV therapy, implying that targeting ICs may optimize antiretroviral therapy in the context of HIV suppression. Here, we systematically review the expression profile, biological regulation, and therapeutic efficacy of targeted immune checkpoints in HIV/SIV infection.

## 1. Introduction

T cell exhaustion due to constant antigen stimulation results in chronic cellular dysfunction. It occurs during chronic infection or cancer when the same antigens repeatedly stimulate T cells, leading to a dysfunctional state [1]. During the process of chronic T cell exhaustion, T cell reactivation gradually declines along with enhanced expression of immune checkpoint (IC) markers. IC markers, primarily PD-1 (programmed cell death protein-1), CTLA-4 (cytotoxic T-lymphocyte-associated antigen-4), TIGIT (T cell immunoglobulin and ITIM domain), LAG-3 (lymphocyte-activation gene-3), Tim-3 (T cell immunoglobulin and mucin domain-3), CD160, 2B4 (CD244), BTLA (B and T lymphocyte attenuator) and VISTA (V-domain Ig suppressor of T cell activation), interact with their respective ligands to ”symphonically” switch T cell activation into T cell exhaustion [2,3]. By balancing co-stimulatory signals and co-inhibitory signals, IC markers guarantee T cell activation, proliferation, and differentiation in response to antigenic stimulation from invading microbes [4].

How increasing IC markers maintains T cell dysfunction and exhaustion has been extensively investigated in cancer [5,6]; by contrast, the mechanism of how IC markers on T cells affect the disease progression in viral infections, particularly in human immunodeficiency virus (HIV), requires more exploration [2,5,7]. In HIV infection, checkpoint molecules were reported to be overexpressed on both CD4^+^ T cells and CD8^+^ T cells in general [8]. Moreover, the regulation of IC marker expression was correlated with HIV disease progression, as reflected in viral RNA replication, T cell function, and HIV reservoir enrichment. In this review, we systematically compared the expression of T cell checkpoint markers with the stage of HIV infection to assess their potential as therapeutic targets.

## 2. HIV Infection Upregulates the Expression of Immune Checkpoints in T Cells

The expression of ICs on T cells varies according to the stage and the level of host immunity against HIV infection. PD-1 is a critical IC marker that is consistently upregulated during HIV infection. The percentage of PD-1^+^ CD4^+^ T cells in HIV-infected individuals (both long-term nonprogressors and common infectors) has been found to be significantly higher than that in healthy persons [9,10,11,12]. A 10% average increase in PD-1^+^ CD4^+^ T cells was seen in HIV-infected hosts as compared with HIV-naïve controls [10,13]. Among HIV-infected individuals, long-term nonprogressors (LTNPs) displayed a much lower percentage of PD-1^+^ CD4^+^ T cells (6–21%) as compared with viremic patients (14–50%) [9,11,12,14,15,16]. The percentage of PD-1^+^ CD4^+^ T cells decreased to 10–35% in peripheral blood mononuclear cells (PBMCs) from common infectors with antiretroviral therapy (ART) [9,11,12,15,17,18]. Similarly, the proportion of PD-1^+^ CD8^+^ T cells has been shown to be 5–23%, 11–20%, 21–50%, and 10–30% in healthy persons, long-term nonprogressors, viremic patients, and ART-treated HIV-infected patients, respectively [9,10,12,14,15,18,19,20,21,22,23]. These data indicate that HIV infection upregulated PD-1 expression on CD4^+^ and CD8^+^ T cells dependent of ART treatment. Except for PD-1, numerous IC markers including CTLA-4, TIGIT, LAG-3, Tim-3, CD160, 2B4, and VISTA have been found to be regulated during HIV infection in general. Moreover, the majority of these IC markers showed regulatory trends similar to those of PD-1 (data in regular type in Table 1). In contrast, Zheng Zhang et al. found that HIV infection downregulated BTLA expression. It was shown that BTLA was reduced from 85% to 60–75% on CD4^+^ T cells and from 69% to 21–42% on CD8^+^ T cells in ART-naïve, HIV-infected patients as compared with healthy controls [24].

The underlying mechanism of regulation of IC expression mediated by HIV infection is still unknown and needs further investigation. How HIV regulates PD-1/CTLA-4/Tim-3 expression in CD4^+^ T cells is partially understood, as the HIV Nef protein upregulates PD-1 transcription in HIV-infected CD4^+^ T cell by activating p38 MAPK [25]. However, the Nef protein has also been reported to downregulate CTLA-4 expression in HIV-infected CD4^+^ T cell to provide optimal cellular conditions for cell activation and viral replication [26]. Regarding the regulation of Tim-3 by HIV, evidence has shown that the transmembrane domain of the HIV Vpu protein could downregulate Tim-3, while the di-leucine motif (LL164/165) of the HIV Nef protein upregulated Tim-3 expression on the surface of HIV-infected CD4^+^ T cells in vitro [27,28].

Non-human primate animal models that are widely used in studying HIV/SIV disease have shown different results as compared with humans. For example, SIV-infected rhesus macaques have relatively stable expression of PD-1, CTLA-4, TIGIT, LAG-3, and Tim-3 on CD4^+^ and CD8^+^ T cells with little change in expression (data in bold type in Table 1).

**Table 1 viruses-14-00581-t001:** Expression profile of immune checkpoint markers on T cells from HIV/SIV-infected individuals.

IC Marker	Healthy Controls		HIV-Infected Individuals		Long-Term Nonprogressors		Viremic Individuals		ART-Treated HIV-Infected Individuals	
	CD4^+^ T Cells	CD8^+^ T Cells	CD4^+^ T Cells	CD8^+^ T Cells	CD4^+^ T Cells	CD8^+^ T Cells	CD4^+^ T Cells	CD8^+^ T Cells	CD4^+^ T Cells	CD8^+^ T Cells
PD-1	5–20% [9,10,11,12] **RM: 12**% [29]	5–23% [9,10,12,19,21,23,30]	10% [10]	10% [10]	6–21% [9,11,16,31]	11–20% [9,31]	14–50% [9,11,12,14,15,16,31] **RM: 6**% [29]	21–50% [9,12,14,15,19,20,21,23,30,31]	10–35% [9,11,12,15,17,18] **RM: 10**% [29]	10–30% [9,11,12,15,18,21,23,30]
TIGIT	10–18% [10,11,32]	30% [10,32,33] **RM: 20**% [32]	20% [10]	50% [10] **RM: 40**% [32]	18–19% [11,32]	50–57% [32,33]	21–30% [11,32]	65–70% [32,33]	17–20% [11,17,32]	45–60% [32,33]
CTLA-4	1–6% [11,12,34,35] **RM: 4**% [29]	1–2% [12,35]	6%-11% [34,35]	1% [35]	3–7% [11,16,31]	0.67% [31]	2–10% [11,12,16,31] **RM: 2**% [29]	0.37, 10–12% [12]	1–8% [11,12,34] **RM: 1**% [29]	5% [12]
LAG-3	6% [36] **RM: 9**% [36]	5–7% [30,36] **RM: 9**% [36]	N/A	N/A	0.065% [31]	0.047% [31]	0.021,14–48% [14,31,36] **RM: 12**% [36]	0.004,8–32% [14,30,31,36] **RM: 15**% [36]	12% [17]	4% [30]
Tim-3	3–15% [10,37] **RM: 1**% [38]	5–29% [10,37] **RM: 5**% [38]	5% [10]	5% [10]	22% [37]	30% [37]	30–41% [14,31,37] **RM: 2**% [38]	14–59% [14,20,37] **RM: 5**% [38]	0.8% [17]	N/A
CD160	3% [10]	1–10% [10,21,30,39]	12% [10]	20% [10]	N/A	N/A	N/A	15–40% [21,30,39]	1.1% [17]	10–16% [21,30]
2B4	3% [10]	40–57% [10,21,30]	10% [10]	65% [10]	5% [31]	70% [31]	10% [31]	75–85% [21,30,31]	9.5% [17]	60–70% [21,30]
BTLA	85% [24]	69% [24]	N/A	N/A	75% [24]	42% [24]	60% [24]	21% [24]	N/A	N/A
VISTA	3% [10]	5% [10]	8% [10]	10% [10]	N/A	N/A	N/A	N/A	N/A	N/A

**RM**, *rhesus monkey*; ART, active antiretroviral therapy; PD-1, programmed cell death protein-1; TIGIT, T cell immunoglobulin and ITIM domain; CTLA-4, cytotoxic T-lymphocyte-associated antigen-4; LAG-3, lymphocyte-activation gene-3, Tim-3, T cell immunoglobulin and mucin domain-3, BTLA, B and T lymphocyte attenuator; VISTA, V-domain Ig suppressor of T cell activation; N/A, not assessed.

## 3. Immune Checkpoints Are Associated with Progression of Disease in HIV/SIV Infection

IC expression has been found to be highly correlated with disease progression after HIV/SIV infection, involving activation, proliferation, and function of T cells (Figure 1), and formation and persistence of HIV/SIV latency (Figure 2, Table 2).

### 3.1. Correlation between ICs from CD28 Superfamily and Progression of Disease in HIV/SIV Infection

PD-1, CTLA-4, and BTLA, members of the CD28 superfamily, are characterized by an immune receptor tyrosine-based inhibitory motif (ITIM) domain in the cytoplasm; PD-1, CTLA-4, and BTLA bind to PD-L1/2, B7, and herpes virus entry mediator (HVEM), respectively, to regulate progression of disease from HIV/SIV infection via transmission of inhibitory signals. PD-1 expressed on CD4^+^ and CD8^+^ T cells is positively correlated with plasma viral RNA (vRNA) replication and negatively correlated with CD4^+^ T cell count in HIV infection [9,13,15,20,40,41,42]. In HIV-infected children, a high proportion of PD-1^+^ memory CD4^+^ and CD8^+^ T cells was also observed, correlating with vRNA, CD4/CD8 ratio, and immune activation [43]. PD-1 is also highly expressed on HIV-specific CD8^+^ T cells, with significant correlation to plasma viral load and CD4 T cell count [9,19,30,44].

It has been reported that overexpression of PD-1 impaired T cell proliferation and caused overactivation and exhaustion of T cells, ultimately leading to HIV progression [2]. In ART-treated HIV patients, both PD-1^hi^ CD4^+^ T cells and PD-1^hi^ CD8^+^ T cells were negatively regulated by HIV-specific T cell immune responses characterized by reduced IFN-γ secretion in response to HIV Gag and Env peptides [15]. Blockade of the PD-1/PD-L1 pathway, especially simultaneous blocking of PD-L1 and stimulation through CD28, significantly enhanced the proliferation of HIV-specific CD4^+^ T cells in ART-naïve, HIV-infected patients [40,45], indicating a synergistic effect by targeting both the inhibitory and costimulatory pathways [45]. In addition, upregulated PD-1 has been observed in HIV-specific CD8^+^ T cells isolated from typical progressors (TPs) but not in long-term nonprogressors (LTNPs), leading to lower IFN-γ levels from HIV-specific CD8^+^ T cells and impaired proliferation of HIV-specific effector memory CD8^+^ T cells [9]. PD-1^hi^ HIV-specific CD8^+^ T cells were shown to accumulate poorly differentiated phenotypes (CD27^hi^, CD28^lo^, CD57^lo^, CD127^lo^, CCR7^–^ and CD45RA^–^) and showed lower production of IFN-γ, IL-2 and TNF-α in response to HIV peptides [44]. Blockade of the PD-1 pathway enhanced proliferation [9,40,44,46] and increased IFN-γ, IL-2 [23,40], TNF-α, granzyme B, and lymphotoxin-α production by HIV-specific CD8^+^ T cells [44]. Polyfunctional HIV Gag-specific CD8^+^ T cell responses to PD-1 blockade could be improved by metformin therapy [47].

In an SIV-infected macaque model, PD-1 was also found to be an essential checkpoint marker during SIV infection. Low PD-1 expression was found in memory and naive CD8^+^ T cells, whereas high PD-1 levels were seen in SIV-specific CD8^+^ T cells isolated from PBMCs, lymph nodes, spleen, and mucosa, which might be attributed to the differences in TCR response [48]. Proliferation was impaired in SIV-specific PD-1^hi^ CD8^+^ T cells [48]. Blocking PD-1 by recombinant rhesus macaque PD-1 (rMamu-PD-1) or rMamu-PD-1-immunoglobulin G (IgG) on PBMCs isolated from SIV-infected rhesus macaques increased the SIV-specific proliferative responses of both CD4^+^ and CD8^+^ T cells [49].

CTLA-4 expression on CD4^+^ T cells has been shown to be closely correlated with HIV progression, and CTLA-4 inhibited cytokine production and proliferation of T cells in chronic HIV infection. The proportion of CTLA-4^+^ CD4^+^ T cells was inversely correlated with CD4^+^ T cell counts and the CD4/CD8 ratio in both ART-naive and ART-treated HIV-1-infected patients [11,34]. The higher proportion of CTLA-4^+^ CD4^+^ T cells co-expressing high levels of CCR5 and Ki-67 resulted in increased susceptibility to HIV infection [34]. A subset of CTLA-4^+^ memory CD4^+^ T cells was also positively correlated with T cell activation (single and dual CD38 and HLA-DR-expressing memory CD4^+^ T cells) during HIV infection, reconfirming that CTLA-4 is a hallmark of HIV disease [11]. Moreover, expression of CTLA-4 was upregulated in HIV-specific CD4^+^ T cells, but could be downregulated by ART [45,50]. The increase in CTLA-4^+^ HIV-specific CD4^+^ T cells was correlated with higher vRNA levels and lower CD4^+^ T cell counts in PBMCs [45,50]. The blockade of CTLA-4 partially rescued cytokine production and proliferation of CD4^+^ and CD8^+^ T cell subsets, as upregulated expression of IFN-γ, CD40L, IL-2, and TNF-α in both CD4^+^ T cells and CD8^+^ T cells [31], and restored the proliferation of HIV-specific CD4^+^ T cell producing IFN-γ and IL-2 [50].

The proportion of BTLA^low^ T cells has been shown to be positively correlated with HIV disease progression of HIV infection. Lower expression of BTLA on both T cells and HIV-specific CD8^+^ T cells was commonly found in HIV-infected patients either as TP or with AIDS, but rarely found in LTNP [24]. The proportion of BTLA^+^ CD4^+^ T cells was positively correlated with CD4^+^ T cell count, yet inversely correlated with vRNA and Ki-67 expression, respectively. Moreover, BTLA-mediated inhibitory signaling of CD4^+^ T cell activation as well as cytokine production was severely impaired in HIV-infected patients [24].

### 3.2. Correlation between ICs from CD4 Family Members and Disease Progression in HIV Infection

LAG-3, a member of the CD4 family, promotes T cell exhaustion by ligating to MHC II, and participates in HIV pathogenesis as a new predictor for disease progression of HIV infection. Tian et al. showed that the numbers of LAG-3^+^ CD4^+^ T cells and LAG-3^+^ CD8^+^ T cells were inversely correlated with CD4^+^ T cell counts, while the number of LAG-3^+^ CD4^+^ T cells was positively correlated with vRNA in HIV-positive individuals [36]. Another study found that the proportion of LAG-3^+^ CD8^+^ T cells was positively correlated with vRNA, but not correlated with CD4 T cell count [20]. A high proportion of LAG-3^+^ CD4^+^ and CD8^+^ T cells was also found to be related to continuous T cell activation, as indicated by co-expression with CD38 in HIV-1-infected individuals [36]. The putative correlation between LAG-3 expression and clinical indicators needs more data to support it.

The increased inhibitory signals transduced by LAG-3 have been shown to be related to functional T cell impairment, including enhanced production of IFN-γ, IL-2, IL-10, and TNF-α in HIV-1-infected subjects in the presence of a LAG-3 blocker, and increased HIV Gag-stimulated proliferation of HIV-specific IFN-γ^+^ CD8^+^ T and IFN-γ^+^ CD4^+^ T cells. Overexpression of LAG-3 consistently validated the inhibitory regulation of LAG-3 during HIV infection. The synergistic enhancement between the LAG-3/MHC II pathway and the PD-1/PD-L1 pathway strongly promoted the T cell-mediated immune response against HIV [36].

### 3.3. Correlation between TIGIT and Disease Progression in HIV/SIV Infection

TIGIT is a common immune receptor linked to PVR/Nectin2 that induces potent immunosuppression, but the correlation between TIGIT expression and HIV infection is controversial. Glen et al. showed that the number of TIGIT^+^ CD8^+^ T cells and TIGIT^+^ CD4^+^ T cells was increased in HIV-infected patients during all stages of AIDS [32]. However, the frequency of TIGIT^+^ CD8^+^ T cells and TIGIT^+^ CD4^+^ T cells was also increased in ART-naïve and long-term ART-treated patients with HIV infection as compared with elite controllers and healthy subjects [11,33]. In HIV-infected patients, the proportion of TIGIT^+^ CD8^+^ T cells was inversely correlated with the CD4 T cell count and the CD4/CD8 ratio [32,33]. Expression of TIGIT on T cells may be related to HIV pathogenesis by mediating HIV replication and host immune activation during HIV infection [32]. Among long-term ART-treated, HIV-infected patients, a positive correlation between the numbers of TIGIT^+^ CD4^+^ T cells and CD4^+^ T cell-associated HIV DNA was found. The frequencies of TIGIT^+^ CD8^+^ T cells and TIGIT^+^ CD4^+^ T cells were positively correlated with the frequency of CD38^+^ HLA-DR^+^ cells in HIV-infected individuals [11,32,33]. In vitro studies indicated that TIGIT might limit CD8^+^ T cell proliferation and function against HIV by a unique pathway [32,33]. However, a recent study proposed that TIGIT upregulated genes associated with antiviral immunity in CD8^+^ T cells, suggesting that TIGIT^+^ CD8^+^ T cells maintain an intrinsic cytotoxicity rather than succumbing to a state of immune exhaustion in HIV-infected individuals [51]. Rhesus TIGIT (rhTIGIT) partially mimics human TIGIT expression and function during HIV infection. In SIV-infected rhesus macaques, expression of rhTIGIT was significantly increased on CD8^+^ T cells derived from lymph nodes (LNs) and spleen but not PBMCs significantly [32]. RhTIGIT inhibited the proliferation and cytokine secretion of SIV-specific CD8^+^ T cells [32].

### 3.4. Correlation between Tim-3 and Disease Progression in HIV/SIV Infection

The numbers of Tim-3^+^ CD4^+^ and CD8^+^ T cells have been shown to be elevated in individuals with acute and chronic progression of HIV-1 infection, but not in patients with viral control, as compared with noninfected individuals. Expression of Tim-3 was upregulated on HIV-specific CD4^+^ and CD8^+^ T cells from HIV-1-infected patients with chronic progression [37,45,52], but this could be effectively reduced by ART. However, no significant increase in Tim-3 expression was found in an HIV-2 cohort [53]. As a potential predictor of HIV-1 disease progression, the expression of Tim-3 on CD4^+^ and CD8^+^ T cells correlated positively with vRNA and CD38 expression, but was inversely correlated with CD4^+^ T cell counts in ART-free HIV infectors [37]. ART strongly attenuated the positive correlation between Tim-3 expression on CD8^+^ T cells, vRNA, and CD4^+^ T cell count [37]; but, the significant association between Tim-3 and CD38 was unaffected by ART [37]. These linear correlations were also observed between plasma HIV viral load and expression of Tim-3 on HIV-specific CD4^+^ T cells [45].

Tim-3-expressing T cells are dysfunctional because of their low capacity to proliferate, produce cytokines, and degranulate in response to ligation of Tim-3 (galectin-9). In both HIV-1 infected patients and healthy individuals, Tim-3 was shown to moderately inhibit IFN-γ, TNF-α, and CD107a production in total and HIV-specific Tim-3^+^ CD4^+^ T cells and Tim-3^+^ CD8^+^ T cells [37,54]. In chronic persistent HIV infection, the proliferation of Tim-3^low^ CD8^+^ T cells was higher than that of their Tim-3^hi^ counterparts [37]. Blocking Tim-3 enhanced proliferation and cytotoxicity in both HIV-specific CD4^+^ and CD8^+^ T cells [37,46,54]. Recently, Tim-3 was found to be a natural immune inhibitor against viral spread [27].

In SIV-infected rhesus monkeys, the percentage of Tim-3^+^ CD8^+^ T cells was significantly higher in LNs than in PBMCs [38]. Amancha et al. found that Tim-3 expression on CD4^+^ and CD8^+^ T cells followed a ”reversed U” curve in PBMCs during SIV infection [55]. Therefore, there was little linear correlation between PBMC/LN-derived Tim-3^+^ T cells and SIV viremia [38,55]. Interestingly, using Tim-3^+^/PD-1^+^ double markers, the expression of these dual ICs on LN-derived CD8^+^ T cells was positively correlated with SIV plasma viremia [38]. The mechanism of Tim-3 regulation of HIV-infected T cells remains controversial and further studies are needed to clarify the pathway [38,55].

### 3.5. Correlation between CD160 and Disease Progression in HIV/SIV Infection

The expression of CD160 has been shown to be increased in CD8^+^ T cells in ART-naïve HIV-1-infected individuals [39,56], which could be remarkably reduced by ART, particularly in naïve CD8^+^ T cells [21,30,56]. Unexpectedly, the subset of CD160^+^ CD8^+^ T cells was higher in HIV-1-infected slow progressors (SPs) and ECs as compared with TPs. These results differed from those obtained for PD-1^+^ and CTLA-4^+^ CD8^+^ T cells in HIV infection [39,57]. Expression of CD160 on CD8^+^ T cells was associated with slow progression of HIV infection, and showed a positive correlation with CD4 T cell counts, but a negative correlation with vRNA. It was concluded that enhanced CD160 expression on CD8^+^ T cells promoted cytotoxicity against HIV [39]. HIV-specific CD160^+^ CD8^+^ T cells were dominantly proliferative and highly activated in ART-naïve, HIV-infected individuals, with increasing expression of granzyme B and CD107a [39]. In contrast, CD160 signaling could attenuate CD8^+^ T cell immunity against HIV chronic infection, largely through immunosuppression of CD160-HVEM (herpes virus entry mediator). Blockade of the CD160/HVEM pathway restored proliferation and cytokine production from HIV-specific CD8^+^ T cells [56]. Moreover, HIV-specific CD8^+^ T cells co-expressing PD-1, CD160, or 2B4, which have been defined as exhausted phenotypes, presented poor activation, reduced intracellular cytokine production, and attenuated responses [30,56,57].

The expression of 2B4 on CD8^+^ T cells has been shown to be higher in ART-naïve HIV patients, and the expression could be reduced by ART [21,30,57]. The cross-linkage of 2B4 with CD48 was considered an inhibitory signal, which downregulated the functionality of CD4^+^ and CD8^+^ T cells during disease progression associated with HIV infection. T cells against HIV could be cooperatively attenuated by co-expression of 2B4 with PD-1 and/or CD160, which was inversely correlated with production of IFN-γ, IL-2, and TNF-α from HIV-specific CD8^+^ T cells [30]. The percentage of HIV-specific CD8^+^ T cells expressing both 2B4 and PD-1 was negatively correlated with perforin production [57]. Blockade of 2B4 elevated the CD40L responses on both CD4^+^ and CD8^+^ T cells, which could be exponentially augmented by PD-1 co-blockade in HIV-positive patients [31]. Co-blockade of PD-1/PD-L1 and 2B4/CD48 synergistically restored the proliferation of HIV-specific CD8^+^ T cells [30]. The findings indicated that 2B4, PD-1, and/or CD160 cooperatively exhausted CD8^+^ T cells against chronic HIV stimulation.

The expression of VISTA was significantly higher on both CD4^+^ and CD8^+^ T cells, with attenuated IFN-γ and TNF-α production in HIV-infected individuals [10].

### 3.6. Correlation between Co-Expression of Checkpoint Markers and Disease Progression in HIV Infection

PD-1 is an essential immune checkpoint marker, involved in multiple synergistic co-functions related to HIV-specific immune regulation. Numerous co-expressing phenotypes of checkpoint markers including PD-1^+^ TIGIT^+^ CD8^+^ T cells, PD-1^+^ TIGIT^+^ CD160^+^ 2B4^+^ CD8^+^ T cells, PD-1^+^ 2B4^+^ CD160^+^ Tim-3^−^ CD8^+^ T cells, and PD-1^+^ TIGIT^+^ CTLA-4^+^ CD4^+^ T cells have been shown to be significantly increased in HIV-infected patients [21,22,32,33]. CTLA-4 was linked to PD-1 expression on HIV-specific CD4^+^ T cells [45,50]. HIV-specific CD4^+^ T cells co-expressing PD-1 and CTLA-4 were found at much higher percentages in viremic patients as compared with ECs [45,50]. Co-expressed checkpoints exhibited stronger correlations with HIV viral load and CD4 T cell counts. The percentage of memory CD4^+^ T cells co-expressing PD-1, CTLA-4, and TIGIT was robustly correlated with CD4 T cell counts, vRNA, CD4/CD8 ratio, and CD4^+^ T cell activation during HIV infection [11]. In addition, PD-1^+^ TIGIT^+^ CD8^+^ T cells were inversely correlated with CD4 T cell counts, but positively correlated with vRNA [32]. HIV-specific CD4^+^ T cells co-expressing PD-1, CTLA-4, and Tim-3 exhibited greater aggregate correlation with vRNA [45]. A modest positive correlation between the percentage of PD-1^+^ CD160^+^ 2B4^+^ LAG-3^−^ HIV-specific CD8^+^ T cells and virus load was shown during chronic HIV infection [30]. In T cells from HIV-viremic individuals, multiple expression of checkpoint markers including PD-1^+^ CD160^+^ CD8^+^ T cells, PD-1^+^ CD160^+^ 2B4^+^ CD8^+^ T cells, PD-1^+^ TIGIT^+^ CD8^+^ T cells, and PD-1^+^ Tim-3^+^ T cells were found to be associated with lower expression of cytokines against HIV as compared with T cells expressing single checkpoint markers [30,32,52,56]. In particular, PD-1^+^ CD160^+^ double pathways co-enhanced in CD8^+^ T cells resulted in poor survival [56]. The in vitro response of HIV-specific CD8^+^ T cells was better restored via blockade of PD-1/PD-L1 synergizing with TIGIT/CD155, 2B4/CD48, Tim-3/galectin-9, or BTLA/HVEM as compared with that from single blockade of PD-1/PD-L1 [30,32,46,52]. HIV-specific CD4^+^ T cell function was improved through co-blockade of PD-L1 with Tim-3 [52]. In summary, the synergistic effects of co-inhibition of these checkpoint markers were remarkable, but more investigation is warranted.

### 3.7. Correlation between Immune Checkpoint Markers and HIV/SIV Latency

HIV reservoirs remain largely understudied. Accumulating proof has shown that checkpoint markers play fundamental roles in maintaining HIV latency (Figure 2). The baseline levels of the checkpoint markers PD-1, Tim-3, and LAG-3 have been shown to be linearly correlated with total baseline HIV-1 DNA, which was predictive of the HIV DNA reservoir after ART, as well as time to rebound after ART withdrawal [14,58]. Such a correlation indicated that checkpoint markers represented the exhaustive dynamics of T cells interacting with HIV reservoirs. Expression of PD-1, TIGIT, and LAG-3 on CD4^+^ T cells was assumed to be positively correlated with the number of persistently infected CD4^+^ T cells considered to be the essential part of the HIV reservoir in patients controlled with successful ART [17,59,60]. PD-1 expressed on lymph node CD4^+^ T cells was characterized as a major source of replication-competent HIV, which presents a serious dilemma for achieving a functional cure [61,62,63].

In SIV-infected rhesus macaques, CTLA-4^+^ PD-1^−^ memory CD4^+^ T cells were found with a large majority of cell-associated SIV Gag DNA in PBMCs, lymph nodes, spleen, and gut tissues in monkeys successfully controlled with ART. CTLA-4^+^ PD-1^−^ memory CD4^+^ T cells isolated from lymph nodes were considered to be a major part of the persistent viral reservoir [64]. In contrast, we recently reported that LP-98 (a HIV fusion-inhibitory lipopeptide) monotherapy showed extremely potent antiviral efficacy in SHIV_SF162P3_-infected rhesus macaques. Interestingly, we found that a higher proportion of PD-1^+^ resting CD4^+^ central memory T cells was measured on both superficial lymph nodes and deep lymph nodes of stable virologic rebound (SVR) macaques as compared with that of stable virologic control (SVC) after withdrawal from LP-98 treatment, highlighting the close relationship between PD-1 and the LN viral reservoir [65]. In HIV-infected humanized mice, PD-1^+^ CD4^+^ T cells and TIGIT^+^ CD4^+^ T cells originating from spleens were enriched with latent and reactivatable HIV-1 [66]. In vitro experiments also indicated that HIV latency was enriched in PD-1, CTLA-4, Tim-3, or BTLA expressing CD4^+^ T cells, which could also be reversed by blockade of ICs [67,68,69,70]. The biological effects of ICs on HIV latency have been investigated with various experimental models, but need to be further verified, and the precise conclusions should be summarized for clinical application.

**Table 2 viruses-14-00581-t002:** Biological role of immune checkpoint markers on T cell function in HIV/SIV infection.

IC Marker	Function Mediated by HIV/SIV Infection			
	CD4^+^ T Cell		CD8^+^ T Cell	
PD-1	Proliferation	Impairs proliferation of HIV-specific CD4^+^ T cells [2,40,45]. Impairs proliferation of SIV-specific CD4^+^ T cells [49].	Proliferation	Impairs proliferation of HIV-specific effector memory CD8+ T cells and HIV-specific CD8+ T cells [2,9,40,44,46]. Impairs proliferation of SIV-specific CD8+ T cells [48,49].
	Cytokine secretion	Reduces IFN-γ secretion by HIV-specific CD4+ T cells [15].	Cytokine secretion	Reduces IFN-γ, IL-2, TNF-α, granzyme B, and lymphotoxin-α secretion by HIV-specific CD8^+^ T cells [9,15,44].
	Virus reservoirs	Contributes to HIV reservoirs in CD4^+^ T cells [14,17,59,60,61,62,63,67,68,69]. Contributes to SIV reservoirs in resting CD4^+^ central memory T cells in LNs [65].		
TIGIT	Virus reservoirs	Contributes to HIV reservoirs in CD4^+^ T cells [17,59,60,68].	Proliferation	Impairs proliferation of HIV-specific CD8^+^ T cell [32]. Inhibits the proliferation of SIV-specific CD8^+^ T cells [32].
			Cytokine secretion	Reduces IFN-γ, TNF-α, and IL-2 production in HIV-specific CD8^+^ T cells [32]. Increases perforin and granzyme B production in CD8^+^ T cells [32]. Decreases the frequency of polyfunctional CD107a^+^GrzB^+^IFN-γ^+^TNF^+^ as well CD107a^−^GrzB^−^IFN-γ^+^TNF^−^ HIV-specific CD8^+^ T cells [33]. Upregulates genes associated with antiviral immunity in CD8^+^ T cells [51]. Reduces IFN-γ secretion by SIV-specific CD8^+^ T cells [32].
CTLA-4	Proliferation	Reduces the proliferation of HIV-specific CD4^+^ T cell producing IFN-γ and IL-2 [50].	Cytokine secretion	Reduces IFN-γ, CD40L, IL-2, and TNF-α by CD8^+^ T cells [31].
	Cytokine secretion	Reduces IFN-γ, CD40L, IL-2 and TNF-α by CD4^+^ T cells [31].		
	Virus reservoirs	Contributes to SIV reservoirs in memory CD4^+^ T cells [64,68].		
LAG-3	Proliferation	Impairs HIV-specific CD4^+^ T cell proliferation [36].	Proliferation	Impairs HIV-specific CD8+ T cell proliferation [36].
	Cytokine secretion	Reduces IFN-γ secretion by HIV-specific CD4^+^ T cell [36].	Cytokine secretion	Reduces IFN-γ secretion by HIV-specific CD8^+^ T cell [36].
	Virus reservoirs	Contributes to HIV reservoirs in CD4^+^ T cells [14,17,59].		
Tim-3	Proliferation	Impairs proliferation of HIV-specific CD4^+^ T cells [37,46].	Proliferation	Impairs proliferation of CD8^+^ T cells and HIV-specific CD8^+^ T cells [37,38,46].
	Cytokine secretion	Inhibits IFN-γ, TNF-α, and CD107a production in total and HIV-specific CD4^+^ T cells [37,54]. Increases IFN-γ, TNF-α, IL-2, and CD107a production in SIV-specific CD4^+^ T cells [55].	Cytokine secretion	Inhibits IFN-γ, TNF-α, and CD107a production in total and HIV-specific CD8^+^ T cells [37,54]. Decreases ability to release perforin and granzymes by HIV-specific CD8^+^ T cells [54]. Inhibits IFN-γ production in total and SIV-specific CD8^+^ T cells [38]. Increases IFN-γ, TNF-α, IL-2, and CD107a production in SIV-specific CD8^+^ T cells [55].
	Virus reservoirs	Contributes to HIV reservoirs in CD4^+^ T cells [14,58,68].		
CD160	N/A		Proliferation	Impairs proliferation of HIV-specific CD8^+^ T cells [56]. Enhances proliferation of HIV-specific CD8^+^ T cells [39].
			Cytokine secretion	Inhibits IFN-γ, TNF-α, and CD107a production in HIV-specific CD8^+^ T cells [56]. Increases CD107a and granzyme B expression in HIV-specific CD8^+^ T cells [39].
2B4	Cytokine secretion	Reduces CD40L production in CD4^+^ T cells [31].	Proliferation	Impairs proliferation of HIV-specific CD8^+^ T cells [30].
			Cytokine secretion	Reduces IFN-γ, TNF-α, IL-2, and perforin in HIV-specific CD8^+^ T cells [30,57]. Reduces CD40L in CD8^+^ T cells [31].
BTLA	Cytokine secretion	Impairs BTLA-mediated suppression of IL-2 and IFN-γ production in CD4^+^ T cells [24].	N/A	
	Virus reservoirs	Contributes to HIV reservoirs in CD4^+^ T cells [68].		
VISTA	Cytokine secretion	Attenuates IFN-γ and TNF-α production in CD4^+^ T cells [10].	Cytokine secretion	Attenuates IFN-γ and TNF-α production in CD8^+^ T cells [10].

## 4. Therapeutic Effects of Immune Checkpoint Blockers on HIV/SIV Infection

Immune checkpoint blockers (ICBs) targeting PD-1 and CTLA-4 pathways combined with anti-HIV drugs have been tested in clinical and experimental trials to develop an effective therapeutic regimen (Table 3 and Table 4).

### 4.1. Immune Checkpoint Blockers Confer Partial Protection against Progression of HIV Infection

The efficacy and safety of ICBs against HIV infection have been reported to vary with pathological conditions [7,71,72,73,74,75,76] (Table 3). In HIV-infected patients with metastatic melanoma, a PD-1 blocker alone or combined with CTLA-4 blocker enhanced the function of HIV-1-specific CD8^+^ T cells and reversed HIV latency in CD4^+^ T cells. The CTLA-4 blocker increased CD4 or CD8 T cell counts, elevated the level of CD4^+^ T cell activation, and reversed HIV latency following drug infusions [67,77,78,79]. In HIV-infected patients with non-small cell lung cancer (NSCLC), a PD-1 blocker helped to restore HIV-specific CD8^+^ T cell function and maintain HIV viremia. However, the PD-1 blocker could not consistently improve CD4 T cell counts and reduce HIV reservoirs [80,81,82,83]. In ART-treated HIV-infected patients with other advanced-stage malignancies, the HIV suppression was inconsistent when used with PD-1 blocker alone [79,84,85]. It is worth noting that anti-PD-1 and anti-CTLA-4 combination therapy moderately reversed HIV latency and potentially eliminated the HIV reservoir in ART-suppressed people living with HIV with advanced malignancies [86]. Moreover, ICBs, including pembrolizumab, nivolumab, ipilimumab, and durvalumab caused no serious adverse events (AEs) to patients, suggesting that patients with HIV should no longer be excluded from clinical trials testing checkpoint inhibitor therapies for various cancers [67,78,80,81,82,83,84,85,87,88,89,90,91,92,93,94]. In HIV-1 infected patients without malignancies, the PD-L1 blocker was shown to enhance the HIV-1 Gag-specific CD8^+^ T cell response [89]. The CTLA-4 blocker reduced plasma viremia in proportion to drug concentration [88]. Safety is a significant challenge, as indicated by the immune-related adverse events (irAEs) that appeared in two clinical trials using a PD-1 blocker for HIV-infected individuals without malignancies [89,95]. Multiple injections of low doses of IC blocker might alleviate the side effects. At present, the differences in therapeutic regimens and the relatively small number of clinical cases might be the cause of the inconsistent outcomes. The tolerance and efficacy of IC blockers vary according to physiological state for HIV infectors with or without malignancies. Given that only clinical trial data for PD-1 and CTLA-4 blockers are available, the data from testing blockers against other ICs (TIGIT, LAG-3, Tim-3, CD160, 2B4, BTLA, and VISTA) may prove highly useful in treatment of AIDs. Therefore, how to combine specific IC blockers and other treatment strategies to design an effective therapeutic program for HIV infectors with or without malignancies is a problem worth studying. More tests of drug combinations in vitro, ex vivo, and in vivo are necessary, to provide the missing information on clinical efficacy for curing AIDS.

**Table 3 viruses-14-00581-t003:** Summary of therapeutic effects of IC blockers in HIV-infected patients.

Reference	IC blocker	Target	Objective	Treatment	Outcomes
Oscar Blanch-Lombarte [78]	Pembrolizumab	PD-1	ART HIV-1-infected individual with metastatic melanoma	Pembrolizumab (2 mg/kg/3 weeks)	Enhances HIV-1 specific-CD8^+^ T cell function. Reduces HIV DNA transiently.
Vanessa A Evans [67]	Nivolumab	PD-1	ART HIV-infected individual with metastatic melanoma	Single intravenous infusion of nivolumab (3 mg/kg)	Increases cell-associated HIV RNA in CD4^+^ T cells. No change of HIV DNA or plasma HIV RNA.
Jillian S.Y. Lau [79]	Nivolumab Ipilimumab	PD-1 CTLA-4	ART HIV-infected individual with metastatic melanoma	Ipilimumab (1 mg/kg/3 weeks) and Nivolumab (3 mg/kg/3 weeks)	Increases cell-associated unspliced HIV RNA. Increases HIV-specific central and effector memory CD8^+^ T cell function. No consistent change of HIV DNA or the proportion of cells with inducible MS HIV RNA.
Fiona Wightman [77]	Ipilimumab	CTLA-4	ART HIV-infected individual with metastatic melanoma	Ipilimumab (3 mg/kg, four doses/3 week)	No change in plasma HIV RNA overall. Decreases plasma HIV RNA, increases CD4 T cell/CD8 T cell counts and enhances CD4^+^ T cell activation cyclically following each ipilimumab infusion. Increases cell-associated unspliced HIV RNA following the first and second infusions.
A Guihot [80]	Nivolumab	PD-1	ART HIV-infected individual with NSCLC	Nivolumab (15 injections/14 days)	Increases plasma HIV RNA transiently. Restores HIV Nef specific CD8^+^ T cells function. Diminishes HIV reservoir.
M Hentrich [81]	Nivolumab	PD-1	ART HIV-infected individual with NSCLC	Chemoradiotherapy and surgical resection Nivolumab (3 mg/kg)	Decreases CD4 T cell counts. HIV-RNA remains undetectable. Slows progressive disease.
Brennan McCullar [82]	Nivolumab	PD-1	ART HIV-infected individual with NSCLC	One cycle of carboplatin/paclitaxel Definitive chemo-radiation with cisplatin and etoposide Start nivolumab	CD4 T cell counts remain stable. Viral load remains undetectable.
Gwenaëlle Le Garff [83]	Nivolumab	PD-1	ART HIV-infected individual with NSCLC	Decompressive radiotherapy Six cisplatin/gemcitabine and four Taxotere chemotherapy treatments Start nivolumab	HIV viral load remains undetectable. CD4 T cell and CD8 T cell counts transiently increased. HIV-specific IFN-γ^+^ CD8^+^ T cells slightly increased. Almost no change in HIV replication or reservoirs.
E P Scully [84]	Nivolumab Pembrolizumab	PD-1	ART HIV-1-infected individuals with malignancies	Nivolumab (participant 1 with head and neck SCC, standard dosing, for 18 months) Nivolumab (participant 2 with head and neck SCC, four doses) Pembrolizumab (participant 3 with squamous cell carcinoma of the skin)	No consistent change in CD4 T cell counts or CD4^+^ T cell cell-associated HIV-1 RNA/DNA or plasma virus load. No change in HIV-1 Gag specific T cell responses.
Neil J Shah [85]	Nivolumab, Pembrolizumab, Atezolizumab, Durvalumab and Avelumab	PD-1/PD-L1	HIV-infected individuals with advanced-stage cancers	Anti-PD-(L)1 monotherapy or anti-PD-(L)1 monotherapy combined with chemotherapy	No change in HIV viral load or CD4 T cell counts. Toxicity and efficacy rates are similar to those observed in patients without HIV.
Thomas A. Rasmussen [86]	Nivolumab Ipilimumab	PD-1 CTLA-4	ART HIV-infected individual with advanced malignancies	Nivolumab (240 mg every 2 weeks) in combination with ipilimumab (1 mg/kg every 6 weeks)	Increases cell-associated unspliced HIV RNA. Eliminates cells containing replication-competent HIV.
Cynthia L Gay [89]	BMS-936559	PD-L1	ART HIV-1-infected adults	Single infusions of BMS-936559 (0.3 mg/kg)	Increases HIV-1 Gag-specific CD8^+^ responses. CD4 T cell counts remain unchanged. No consistent change in HIV-1 DNA or RNA/DNA ratio.
Elizabeth Colston [88]	Ipilimumab	CTLA-4	Chronic HIV-1 -infected individuals	Ipilimumab, 0.1, 1, or 3 mg/kg, two doses every 28 days; or 5 mg/kg, four doses every 28 days	Decreases plasma HIV-1 RNA of two patients treated with 0.1 mg/kg and 1 mg/kg ipilimumab. Increases plasma HIV-1 RNA in 14 patients treated with 3 or 5 mg/kg ipilimumab. No change in CD4 and CD8 T cell counts.

### 4.2. Immune Checkpoint Blockers Confer Partial Protection against Progression of SIV Infection

SIV-infected rhesus macaques are the most widely used nonhuman primate (NHP) models for exploring the molecular mechanism and the potential application of novel HIV therapies. Analysis of ICB outcomes in NHP models has provided an experimental foundation and theoretical basis for HIV clinical research (Table 4). In SIV-infected rhesus macaques, the PD-1 blocker enhanced proliferation and function of SIV-specific CD4^+^ and CD8^+^ T cells, as well as increased the proliferation of memory B cells with elevated SIV Env-specific antibody [96,97,98,99]. A PD-1 blocker together with ART reduced viremia to a low level, increased the CD8^+^ T cell response, and in general, decreased the rebound after ART withdrawal [98,100,101,102]. The CTLA-4 blocker decreased vRNA levels in lymph nodes and restored the effector function of SIV-specific CD4^+^ and CD8^+^ T cells during ART without impeding rebound after ART suspension [103]. However, Justin Harper et al. recently reported that the combination of PD-1 blocker and CTLA-4 blocker was more effective than single blockade alone in robust latency reversal, while neither PD-1 blocker nor CTLA blocker could enhance SIV-specific CD8^+^ T cell function [104].

Using a prophylactic model with PD-1 blocker, the vaccine-induced, SIV-specific CD8^+^ T cell and B cell response has been augmented, resulting in better control of pathogenic SIV infection [105,106,107]. However, PD-1 blockade combined with therapeutic vaccination might be a double-edged sword for SIV therapy. In chronically SIV-infected rhesus macaques under ART, Sheikh et al. discovered that a PD-1 blocker could be used to enhance the therapeutic effects of SIV vaccination by improving vaccine-induced CD8^+^ T cell function, sustaining B cell homing into follicles for vaccine-induced CD8^+^ T cell responses, reducing the viral reservoir in lymphoid tissue and, thereby, controlling HIV rebound upon ART interruption [108]. Surprisingly, Wu et al. found the opposite outcome: PD-1 blockade in combination with therapeutic vaccination led to viral reservoir activation that accelerated viral rebound in chronically SIV-infected macaques after ART interruption [109]. Therefore, the therapeutic potential of vaccine + PD-1 blockade might be closely related to ART treatment and the use of this drug combination still needs to be optimized in the SIV-infected rhesus macaque model for developing an HIV-1 cure in the future. Activation of the viral reservoir and enhancement of SIV-specific CD8^+^ T cell functions are the two dominant factors that need to be considered in the design of an immunotherapeutic strategy. Combining IC blockers with vaccine might produce better outcomes for HIV therapy, but may also raise more safety concerns. Much work is required to determine the best combination to control virus without excessive toxicity.

**Table 4 viruses-14-00581-t004:** Summary of therapeutic effects of IC blockers in macaque model.

Reference	IC Blocker	Target	Objective	Treatment	Outcomes
Vijayakumar Velu [96]	Humanized mouse anti-human PD-1 Ab (clone EH12-1540)	PD-1	SIV251/SIVmac239-infected Indian rhesus macaques	Anti-PD-1 Ab (3 mg/kg) in early chronic phase and in late chronic phase on days 0, 3, 7, and 10	Increases SIV-specific CD8^+^ T cells with improved functionality. Increases proliferation of memory B cells and SIV envelope-specific antibodies. Reduces plasma viral load. Prolongs survival of SIV-infected RMs.
Adam C Finnefrock [107]	Anti-human PD-1 Ab (clone 1B8)	PD-1	SIVmac239-infected rhesus macaques	Therapeutic model: single infusion of anti-PD-1 Ab 1B8 (5 mg/kg) to chronic SIV-infected macaques before or during ART Prophylactic model: anti-PD-1 Ab 1B8 (5 mg/kg) to naive macaques immunized with an SIV-Gag adenovirus vector vaccine	Therapeutic model: Transiently increases viral load; PD-1 blockade during ART has no discernible effect on SIV Gag-specific CD8^+^ T cells and CD4 T cell counts. Prophylactic model: Enhances SIV Gag-specific CD8^+^ T responses.
Ravi Dyavar Shetty [97]	Mouse anti- human PD-1 Ab	PD-1	SIV-infected rhesus macaques	Anti-PD-1 Ab (3 mg/kg) at either 10 or 90 weeks after SIV infection on 0, 3, 7, and 10 days	Downregulates type I IFN responses. Enhances gut junction-associated gene expression and reduces microbial translocation. Enhances immunity to pathogenic gut bacteria. Prolongs survival of SIV-infected RMs.
Praveen K Amancha [99]	Recombinant macaque PD-1 fused to macaque Ig-Fc (rPD-1-Fc)	PD-1	SIVmac239-infected rhesus macaques	rPD-1-Fc (50 mg/kg) alone or in combination with ART during the early chronic phase	Enhances SIV-specific CD4^+^ and CD8^+^ T cell proliferation and function. Fails to alter plasma viremia. Induces a significant delay in viral load rebound after ART interruption.
Geetha H Mylvagana [98]	Primatized anti–PD-1 Ab (clone EH12- 2132/2133)	PD-1	Chronic SIVmac251-infected rhesus macaques	Stage I: anti-PD-1 Ab (3 mg/kg/dose, 5 doses) between 24 and 30 weeks after infection on days 0, 3, 7, 10, and 14. Stage II: RMs again treated with anti-PD-1 Ab (10 mg/kg/dose, three monthly, 3 doses) at 26–30 weeks following ART initiation	Improves SIV Gag-specific CD8^+^ T cell functions prior to ART. Reduces the viral reservoir after ART initiation. Enhances viral control after ART interruption.
Diego A Vargas- Inchaustegui [102]	B7-DC-Ig fusion protein	PD-1	Chronic SIVmac251- infected rhesus macaques	ART plus B7-DC-Ig (10 mg/kg, weekly, 11 weeks), then B7-DC-Ig alone for 12 weeks	Maintains lower viremia, favorable T cell/Treg repertoire, and lower SIV-specific responses.
Elena Bekerman [110]	Human/rhesus chimeric anti- PD-1 antibody	PD-1	ART SIVmac251-infected rhesus macaques	Anti-PD-1 chimeric Ab (10 mg/kg, every other week, four doses) with or without TLR7 agonist vesatolimod (0.15 mg/kg, every other week, 10 doses)	No change in viral rebound kinetics following ART interruption. No change in SIV reservoir size. No change in frequency and function of SIV-specific T cells.
Sheikh Abdul Rahman [108]	Primatized anti–PD-1 antibody (clone EH12)	PD-1	Chronical SIVmac239-infected rhesus macaques	Immunized with a CD40L plus TLR7 agonist–adjuvanted DNA/MVA SIV239 vaccine (DNA vaccine: 1 mg/333 µL, 600 µL/dose, at weeks 38 and 42 MVA vaccine: 1 mL/dose, at weeks 46 and 60) during ART. Received anti–PD-1 treatment on days 0, 3, 7, 10, and 14, starting 10 days before the initiation of ART (3 mg/kg) and on week 38–44 starting with the first DNA prime during ART (10 mg/kg, 3 doses, every 3 weeks)	Increases the frequency of cytolytic CD8^+^ T cells in the blood and LN. Enhanced cytolytic CD8^+^ T cells localization in germinal centers of B cell follicles. Reduces viral reservoirs in lymphoid tissue. Controls viral rebound upon ART discontinuation.
ChunxiuWu [109]	Anti-PD-1 antibody (GB226)	PD-1	ChronicallySIV-infected macaque	Anti-PD-1 antibody injection (20 mg/kg) every 2 weeks from 1 to 7 weeks and rAd5-SIVgpe (10^11^vp in 1 mL PBS) at weeks 0 and 4 post ART discontinuation; ART treatment begins at week 3 before the initial vaccination	Exacerbates viral rebound after ART interruption. Accelerates the reactivation of the latent reservoir and AIDS progression. Increased CD4 and CD8 T cell counts. Improve SIV-specific CD4^+^ and CD8^+^ T cell immune function.
Enxiang Pan [106]	Genolimzumab	PD-1	Chinese rhesus monkeys	Genolimzumab injection (20 mg/kg, every two weeks) at weeks −1, 1, 3, 5, and 7 and rAd5-SIVgpe (10^11^vp) injection at week 0 and 4; at week 42 after the initial vaccination, animals were challenged with repeated low-dose SIVmac239	Augments and sustains vaccine-induced SIV-specific CD8^+^ T cell responses. Confers better control of pathogenic SIVmac239 infection.
Ping Che [101]	Avelumab	PD-L1	ART SIVmac239-infected rhesus macaques	Avelumab (20 mg/kg, weekly, for 24 weeks) and rhIL-15 (10 µg/kg, daily, continuous infusion for 10 days, two cycles), then, ART was discontinued and avelumab treatment continued until completion of the 24-week treatment	Transiently increases proliferation of natural killer and CD8^+^ T cells. Expands CXCR3^+^PD-1^−/low^ CD8^+^ T cell subset with ability to secrete cytokines. No change in plasma viremia after ART interruption.
Amanda L Gill [100]	Avelumab	PD-L1	ART SIVmac239-infected rhesus macaques	Avelumab (20 mg/kg, weekly) At week 24, all treatments were discontinued	Leads to a trend of transient viral control after discontinuation of treatment.
Anna Hryniewicz [103]	MDX-010	CTLA-4	ART SIVmac251-infected rhesus macaques	Administered MDX-010 (10 mg/kg/injection) after ART initiation at weeks 5 and 8.	No impact on viral rebound following ART suspension. No impact on CD4 T cell and CD8 T cell counts. Decreases viral RNA level in LNs. Increases effector function of SIV-specific CD4^+^ and CD8^+^ T cells.
Todd Bradley [105]	Ipilimumab	CTLA-4	Cynomolgus macaques	Immunized with recombinant CH505 HIV Env gp120 (100 µg every 4 weeks) and ipilimumab (10 mg/kg) during the immunization 1–3	Promotes germinal center activity. Enhances HIV-1 Env antibody responses.
Justin Harper [104]	Nivolumab Ipilimumab	PD-1 CTLA-4	SIVmac239-infected rhesus macaques	Weekly nivolumab and ipilimumab over four weeks during ART, then, ART was interrupted two weeks afterwards with a seven-month follow-up	Enhances T cell proliferation and response in LNs and PBMCs. Induces robust viral reactivation in plasma. Decreases total and intact SIV-DNA in CD4^+^ T cells during ART in LNs. No change in SIV-specific CD8^+^ T cell responses during ART. No control of viremia after ART interruption.

In summary, the blockade of checkpoint targets is a promising strategy for controlling the disease progression of HIV or SIV infection functionally and clinically. More studies are on the way for translating this treatment into new patient regimens, hopefully soon.

## 5. Conclusions

Immune checkpoint markers serve as novel biomarkers for HIV infection. ICs control the switching of T cells between the states of activation and exhaustion, and are closely related to HIV pathogenesis. IC overexpression in CD4^+^ and CD8^+^ T cells attenuated the HIV-specific T cell response. Moreover, ICs are preferentially expressed on the surface of persistently HIV-infected T cells. CD4^+^ T cells overexpressing ICs are commonly found to serve as HIV reservoirs, and the ICs might play an essential role in HIV latency. Correspondingly, the blockade of ICs was shown to reverse HIV latency and restore the T cell function, providing promising therapeutic opportunities for HIV therapy both experimentally and clinically. However, the underlying regulatory mechanism of IC expression triggered by HIV infection and the cellular immune pathways mediated by IC blockers are still unknown and need further investigation. The optimum therapeutic design of ICs blockers alone or combined with other robust antiviral strategies under study offer novel strategies for the achievement of an HIV cure.

## Figures and Tables

**Figure 1 viruses-14-00581-f001:**
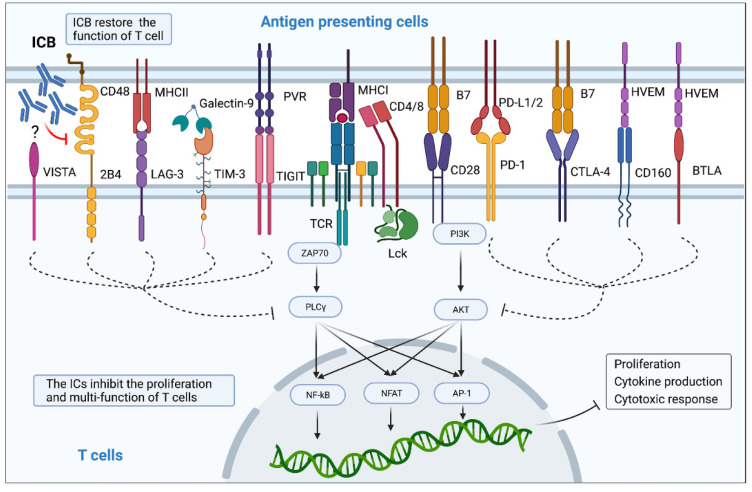
Immune checkpoint proteins impair proliferation and multi-functionality of T cells. Interaction between immune checkpoint markers and their ligands transmits inhibitory signals to CD4^+^ and CD8^+^ T cells. These inhibitory signals limit TCR- and CD28-mediated co-stimulatory pathway, including the NF-κB, NFAT, and AP-1 signaling pathways, which suppress proliferation, cytokine production, and cytotoxic response. IC blockers are able to cause ligand dissociation from the ICs and restore T cell activation.

**Figure 2 viruses-14-00581-f002:**
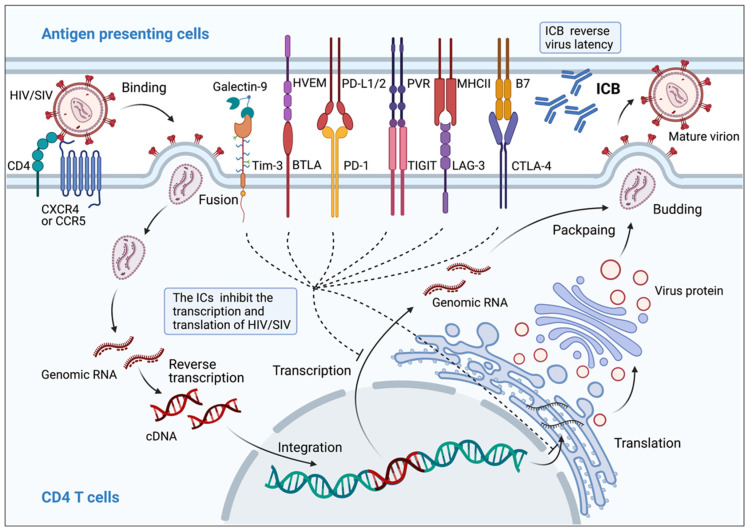
Immune checkpoint proteins contribute to HIV/SIV latency. PD-1, TIGIT, LAG-3, CTLA-4, Tim-3, and BTLA signaling pathways are involved in generating an HIV reservoir and its persistence. The binding of these ICs and their ligands enforces silencing of integrated HIV DNA at the transcriptional and/or translational level. IC blockers can reactivate HIV from the cellular reservoir and produce new virions by blocking IC-ligand interactions in CD4^+^ T cells.

## Data Availability

Not applicable.

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
