# Peer review of "Expression Profile and Biological Role of Immune Checkpoints in Disease Progression of HIV/SIV Infection"

_viruses, 2022, doi:10.3390/v14030581_

Round 1
Reviewer 1 Report
Sun and Xue have summarized current knowledge on immune checkpoint (IC) marker expression in HIV/SIV infection. Specifically, the authors have discussed expression of IC markers in vivo and in vitro and a potential of IC markers as a therapeutic target. This is a very interesting and comprehensive review, which provides new insights into the role of IC markers in HIV pathogenesis and therapy. Here are some suggestions and minor concerns.
The authors exhaustively described findings on IC marker expression on T cells in vivo and its role in T cell functions and so on. Since findings are complicated and sometimes controversial, it is still hard to get the general picture. A summary table like Table2 would be very helpful for readers especially regarding PD-1 and CTLA4. Please consider making one.
We now know much about IC marker expressions and their potential roles in T cell function and HIV latency. However, there are many questions to be studied. For example, what determines which marker is expressed; which combinatory expression has the most impact on HIV biology; are IC markers reversible? etc. It would be nice to discuss remaining questions very briefly in Conclusion.
Minor points:
Line 38-39, “independent of ART treatment”. Table 1 shows some reduction in PD-1+ CD4 and CD8 T cell. Aren’t there some effects of ART?
Line 196, the title for the paragraph 3.5 needs to be corrected.
Line 278, please explain briefly what LP-98 is.
Figure 2, square panel “The ICs inhibit the transcription and translation of DNA”. “Translation of DNA” needs to be corrected.
Reviewer 2 Report
Nowadays, blocking Immune checkpoints is popular in tumour therapy. Immune checkpoint markers serve as novel biomarkers for HIV infection. Blockade of ICs was shown to reverse HIV latency and restore the T cell function, providing promising therapeutic effects on HIV therapy both experimentally and clinically. Sun and Xue gave us a comprehensive systematic review about the immune checkpoints expression profile and biological function during HIV/SIV infection, which is inspiring and significant.
Major
1, Please make a brief introduction about how HIV/SIV upregulated CIs markers expression during the infection.
2, Please discuss the cellular immune pathways regulation upon the blockage of ICs.
3, Please discuss the future of immune checkpoint blocking combined HIV vaccine therapy. (https://www.science.org/doi/10.1126/sciimmunol.abh3034)
4, What are the clinical challenges of CIs for HIV therapy and how to optimize them?
Minor
In line 196, the subtitle of "3.5 Correlation between Tim-3 and disease progression in HIV/SIV infection", Tim-3 should be CD160 instead?
